# Ultra-High Sensitivity Ultrasonic Sensor with an Extrinsic All-Polymer Cavity

**DOI:** 10.3390/s22187069

**Published:** 2022-09-19

**Authors:** Zongyu Chen, Bo Dong, Wobin Huang, Yunji Yi, Chichiu Chan, Shuangchen Ruan, Shaoyu Hou

**Affiliations:** College of Integrated Circuits and Optoelectronic Chips, Shenzhen Technology University, Shenzhen 518118, China

**Keywords:** fiber acoustic sensor, all-polymer cavity, ultrasonic sensor

## Abstract

An ultra-high sensitivity ultrasonic sensor with an extrinsic all-polymer cavity is presented. The probe is constructed with a polymer ferrule and a polymer-based reflection diaphragm. A specially designed polymer cover is used to seal the cavity sensor head and apply pretension to the sensing diaphragm. It can be manufactured by a commercial 3D printer with good reproducibility. Due to its all-polymer structure and high coherence depth, the sensitivity of our proposed sensor is improved significantly compared with that of the other sensor structures. Its sensitivity is 189 times as great as that of the commercial standard ultrasonic sensor at the ultrasonic frequency of 50 KHz, and it has a good response to ultrasonic within the frequency range of 18.5 KHz–200 KHz.

## 1. Introduction

A sound wave is a mechanical wave that plays an important role in our society as a carrier of information. Typically, sound waves with a frequency greater than 20KHz are called ultrasonic waves [1,2]. Conventional piezoelectric transducers (PZT) dominated the ultrasonic detection field for decades before the emergence of the fiber acoustic sensor (FAS). PZT has many limitations in its application, making it impossible to use in various special applications. The FAS has attracted much attention for decades since 1977 [3,4]. Since the FAS has the advantages of high sensitivity, high-temperature resistance, chemical resistance, and electromagnetic interference resistance, it has a wide range of applications in both military and civilian fields for ultrasonic signal detection, especially for the applications in underwater acoustic detection; partial discharge detection of high-voltage equipment; ultrasound medicine; nondestructive testing of building structures and equipment; and navigation [5,6,7,8]. Based on the sensing mechanism difference, FAS can be subdivided into fiber Bragg grating (FBG) [9,10,11,12,13], Michelson interferometer (MI) [14,15,16], Mach–Zehnder interferometer (MZI) [17,18,19], Sagnac interferometer (SI) [20,21], and Fabry–Perot interferometer (FPI) [22,23,24,25,26,27,28,29,30] types. For the FBG, its sensitivity is limited due to the high Young’s modulus of the fiber. For the MZI, one of its sensing arms is usually used for ultrasonic sensing and the other one is used as a reference. Although the MZI has a wide response frequency range, the whole sensor will be particularly bulky due to its structure. The sensing principle of the MI is similar to that of the MZI, so it faces the same challenges as the MZI. For the SI, in order to improve the stability of the optical signal, it often requires the introduction of polarization-dependent devices, which leads to its large size and increases the complexity of the SI sensor system. For the conventional FPI, especially the intrinsic Fabry–Perot interferometer ultrasonic sensor, zirconia ceramic, silica, or metal ferrule is usually used as the sensor probe shell, which cannot help to improve the sensitivity of the sensor due to the high Young’s modulus of such a material.

In addition, in previous studies, many materials have been used to make diaphragms, such as silicon or its compounds [31,32], polymers [24,33,34], two-dimensional materials [23,35], and metal [36]. The reflectivity of the silicon or its compounds can be improved by single-sided polishing, but it is difficult to provide high sensitivity because of its high Young’s modulus and large thickness. If only the polymer is used as the diaphragm, the coherence depth of the reflection spectrum of the sensor is lower due to its lower reflectivity. The difficulty of the two-dimensional material diaphragms lies in their complex fabrication process, and their instability is also a challenge. Therefore, it is a feasible scheme to improve the reflectivity of the polymer material diaphragm by depositing the metal reflective film on the polymer material diaphragm.

Here, an all-polymer ultrasonic sensor with ultra-high sensitivity is proposed. A polymer ferrule with a cover is used as the probe shell, and a polymer diaphragm coated with aluminum film is used to construct the reflective diaphragm of the cavity sensor. To keep fine sealability of the sensor head and the good recoverability of the polymer diaphragm, a specially designed polymer cover is used to package the cavity sensor head. Since the all-polymer structure sensor has a lower Young’s modulus with a high coherence depth, when the ultrasonic pressure is applied to the sensor, the diaphragm and the shell will respond simultaneously, which leads to the high sensitivity of this type of sensor. Experimental results show that its sensitivity is about 189 times as great as that of the commercial standard ultrasonic sensor at the ultrasonic frequency of 50 KHz, and it has good response to ultrasonic within the frequency range of 18.5 KHz–200 KHz. This type of sensor can be fabricated using a commercial 3D printer with good reproducibility, it is low cost with a compact structure and is expected to be widely used in the ultrasonic detection industry.

## 2. Principle of the Sensor

The structure of the sensor is shown in Figure 1a,b. It is constructed by the polymer ferrule, polymer cover, polymer diaphragm, and a lead-in single mode fiber (SMF) packaged in the zirconia ferrule. The material of the polymer ferrule and polymer cover is UV curable resin provided by Formlabs while the polymer diaphragm is based on the polyimide material.

The fabrication process of the sensor is as follows: the first step is coating about 200 nm of aluminum film on the polyimide diaphragm by magnetron sputtering. The cover and ferrule of the sensor probe are modeled using SOLIDWORKS, and then the model file (STL) will be processed in 3D printer software, called PreForm, which is used to define print parameters. Then, the file that includes processing parameters will be uploaded to the commercial 3D printer (Formlabs, Form3) to fabricate the entity model, followed by UV curing post-treatment and heat aging. After that, the cover, ferrule, and polyimide film are assembled and fixed with UV adhesive (ergo 8500). The air cavity length of the sensor is adjusted by the six axes micro-displacement platform. At the same time, we observe the spectrum changes via the optical spectrum analyzer (OSA, YOKOGAWA AQ6370D) to obtain a fine reflection spectrum. Finally, the whole sensor is fixed using UV adhesive. A specially designed polymer cover is used to seal the sensor head and pre-tense the reflectance diaphragm. The thickness/effective diameter of the polymer-based diaphragm is about 25 μm/5.7 mm. Figure 2 shows the typical reflection spectrum of the sensor in air or underwater (fresh water), and the coherence depth of the sensor is about 18.5 dB. Its free spectrum range (*FSR*) formula can be given by
(1)$$L=\frac{\lambda_{1}\lambda_{2}}{2n_{1}\left(\lambda_{1}-\lambda_{2}\right)}=\frac{\lambda_{1}\lambda_{2}}{2n_{1}FSR},$$
where $\lambda_{1}$ and $\lambda_{2}$ are the wavelengths of two adjacent peaks in the reflection spectrum respectively, and $n_{1}$ is the air refractive index; hence the cavity length is about 415 μm. Furthermore, the test of its reflection spectrum is conducted under room temperature of about 28 °C.

An ultrasonic wave belongs to a dynamic signal and it needs to be transformed into a detectable physical parameter. The principle of the sensor is to transform the vibration of the ultrasonic signal into the elastic deformation of polyimide diaphragm and polymer shell, which leads to the change of the cavity length and then leads to the change of the interference spectrum. Therefore, the all-polymer structure of the sensor, especially the polyimide diaphragm coated with aluminum film, is the key to converting sound waves to sound pressure deformation. When the sound wave propagates to the diaphragm, the reflectivity of the incident light can be given by:(2)$$R=\frac{I_{R}}{I_{0}}=\frac{R_{1}+R_{2}-2\sqrt{R_{1}R_{2}}\cos\varphi }{1+R_{1}R_{2}-2\sqrt{R_{1}R_{2}}\cos\varphi },$$
where $R_{1}$ and $R_{2}$ are the reflectivities of the end surface of the SMF and the reflection surface of the film, respectively; $I_{0}$ is the incident light intensity and $I_{R}$ is the reflective light intensity; φ is the phase of the sensor. Note that the length of the cavity is affected by the deformation of the polymer based structure and the deflection of the diaphragm simultaneously under the action of sound pressure, hence the cavity length change can be expressed as:(3)$$\Delta L=\xi \Delta L_{f}+\Delta L_{\max},$$
where $\Delta L_{f}$ is the cavity length variation due to the deformation of the polymer based structure caused by the sound pressure, and ξ is the conversion coefficient; $\Delta L_{\max}$ is the maximum deformation at the center of the diaphragm due to the sound pressure and it can be given by:(4)$$\Delta L_{\max}=\frac{(1-\mu )r^{2}}{2EL_{2}}\Delta P,$$
where μ and E are the Poisson’s ratio and Young’s modulus of the diaphragm material, respectively, and r is the effective radius of the diaphragm. Hence, under the sound pressure, the phase of the cavity can be given by:(5)$$\varphi =\frac{4\pi \left(L_{0}+\xi \Delta L_{f}+\frac{(1-\mu )r^{2}}{2{\mathrm{EL}}_{2}}\Delta P\right)}{\lambda },$$
where L is the initial length of the cavity and ΔL is the cavity length change due to the sound pressure. According to Equations (2) and (5), the reflectivity of the sensor is related to both the deformation of the polymer based structure, the deflection of the diaphragm, and the Young’s modulus of the polymer.

We used the Solid Mechanics (solid) module of the COMSOL Multiphysics finite element (FEM) simulation method for the eigenfrequency calculation of the sensor. Setting the Young’s modulus and density of the polyimide as 2.914 GPa and 1390 kg/m^3^, respectively, and the Young’s modulus and density of the sensor shell as 2.7 GPa and 1170 kg/m^3^, respectively. The mesh type setting is physics-controlled mesh, element size is normal. The contact area between the zirconia ferrule and the polymer part is set as the fixed constraint condition. The simulated result shows that the first-order eigenfrequency of the ultrasonic sensor is around 47.85 KHz, as shown in Figure 3. Meanwhile, the different colors of the surface in Figure 3a represent its displacement magnitude, and the surface arrows indicate its displacement field. Figure 3b shows the deformation of the membrane.

## 3. Experimental Result and Discussions

Figure 4 shows the schematic diagram of the experimental setup. A piezoelectric transducer (PZT, LIYU DYW-50/200-NA) with operation frequency of 50 KHz is placed in the water tank as an ultrasonic generator to generate ultrasonic wave, and its driving voltage is controlled by a function generator (UNI-T UTG9010C 10 MHz). The sensor is placed in the tank and is about 10 cm away from the PZT. A commercial standard ultrasonic sensor (ANHUI HONGYUAN RHS-10, −210 dB) is placed next to the sensor as a calibrator to calibrate the sensor. A tunable laser diode (LD, OCLARO TL5000ZCL) operated within C band with an output power of 10 dBm is used to provide single wavelength laser output as the optical source of experiment system. A photoelectric detector (PD, SHOW PDB1008) with 80 MHz bandwidth is used to detect the optical sensing signal. An oscilloscope (OSC, TEKTRONIX TDS2024C) is used to capture electrical sensing signals. During the experiment, the maximum peak-to-peak (PP) output voltages of the sensor and the calibrator sensor both increase quasi-linearly with the increase of the driving voltage. The relationships between the maximum PP voltage and diver voltage of the sensor and the calibrator sensor at 28.5 KHz and 50 KHz are shown in Figure 5a,b, respectively.

At 28.5 KHz and 50 KHz, the sensitivities of the sensor are 112 and 189 times of the ultrasonic sensor, respectively. Since the operation frequency of 50 KHz is near the eigenfrequency of the simulated result, the sensitivity at 50 KHz is about 1.7 times as big as that at 28.5 KHz, which matches our theoretical predication. It should be noted that the output voltage is impacted by the reflection and diffraction of ultrasonic waves in the water tank, so the use of the sound-absorbing materials can improve the measurement accuracy. The test of its sensor sensitivity is conducted under room temperature of about 28 °C.

On the basis of the above experiments, we also tested its ultrasonic frequency response. Figure 6a,b show the ultrasonic frequency response range of the sensor under room temperature. It can be seen that the sensor has good response at ultrasonic frequencies from 18.5 KHz to 200 KHz. At the same time, since the sensor would be used under room temperature, we measure the temperature response of the sensor under the temperature range between 30 °C and 60 °C, as shown in Figure 7a,b. The sensor shows a quasi-linear response to temperature variation within the range of 30–60 °C. The result shows that its temperature sensitivity is about 657 pm/°C. Hence, it is thermally sensitive. When it is used as a practical ultrasonic sensor, a package structure with a fixed temperature is needed. Alloy material-based porous structures will be applied to the package structure in further work.

## 4. Conclusions

In conclusion, an ultra-high sensitivity all-polymer ultrasonic sensor is proposed and demonstrated. Attributed to its all-polymer structure and high coherence depth, the sensor shows a higher sensitivity of 189 times as high as that of the commercial standard ultrasonic sensor, and it shows a wide ultrasonic frequency response. Since it can be fabricated with a commercial 3D printer with good reproducibility, it is convenient for mass production. Moreover, the sensor exhibits the advantages of low cost, small size, compact structure, simple manufacture, and high sensitivity, and it is expected to have potential applications in the ultrasonic measurement industry.

## Figures and Tables

**Figure 1 sensors-22-07069-f001:**
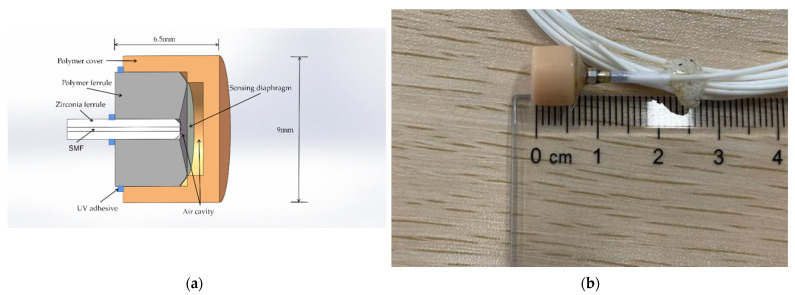
(**a**) Schematic structure of the polymer cavity sensor; (**b**) photograph of the polymer cavity sensor.

**Figure 2 sensors-22-07069-f002:**
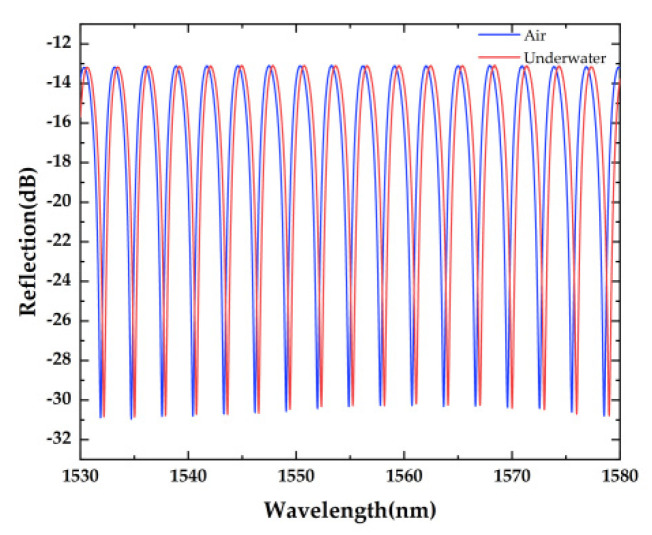
The reflection spectrum of the polymer cavity sensor.

**Figure 3 sensors-22-07069-f003:**
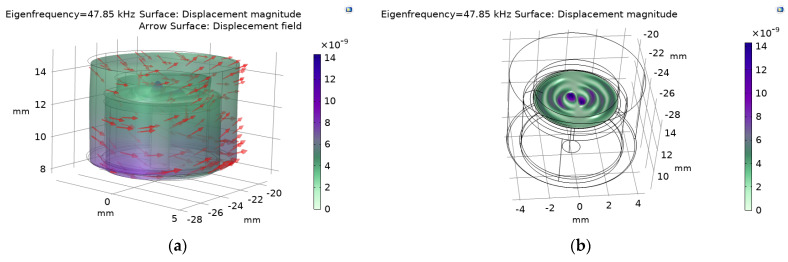
Simulated results of the polymer cavity sensor: (**a**) deformation of the sensor and (**b**) deformation of the membrane.

**Figure 4 sensors-22-07069-f004:**
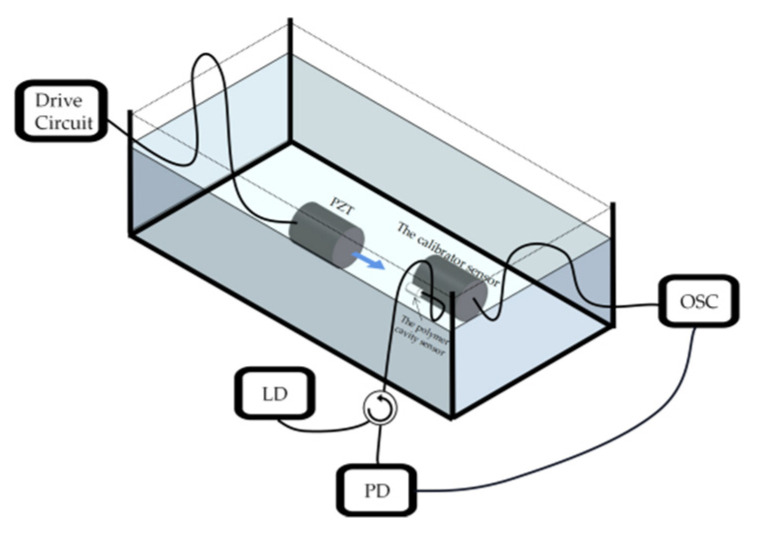
Schematic configuration of the experimental setup.

**Figure 5 sensors-22-07069-f005:**
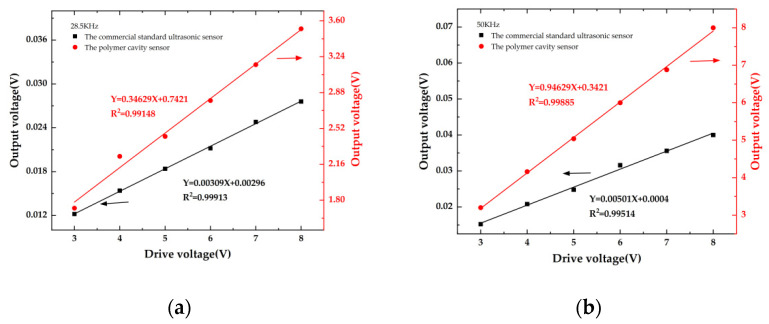
Relationship between the maximum PP voltage and diver voltage of the polymer cavity sensor and the commercial standard ultrasonic sensor at (**a**) 28.5 KHz and (**b**) 50 KHz, respectively.

**Figure 6 sensors-22-07069-f006:**
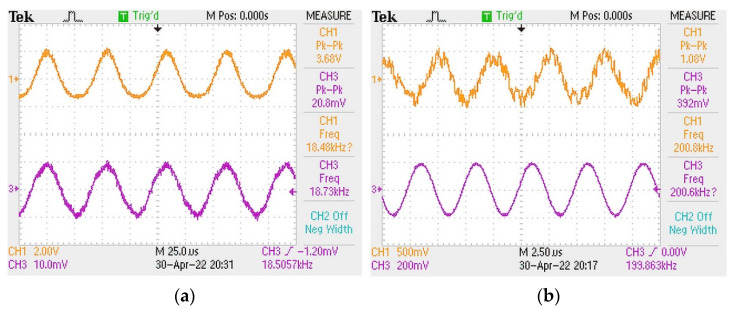
Ultrasound responses at (**a**)18.5 kHz and (**b**) 200 kHz of the sensor and the commercial standard ultrasonic sensor. The yellow curve is the sensor, and the purple curve is the commercial standard ultrasonic sensor.

**Figure 7 sensors-22-07069-f007:**
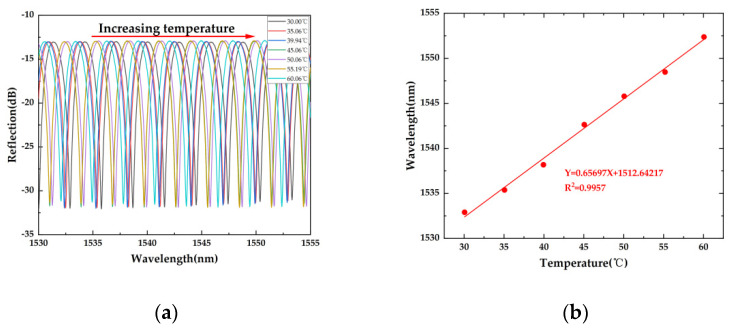
(**a**) Reflection spectrum against temperature of the sensor; (**b**) relationship between the temperature and wavelength shift.

## Data Availability

Not applicable.

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
