# Peer review of "Ultra-High Sensitivity Ultrasonic Sensor with an Extrinsic All-Polymer Cavity"

_sensors, 2022, doi:10.3390/s22187069_

Round 1

Reviewer 1 Report

The authors demonstrated an extrinsic all-polymer cavity ultrasonic sensor. Its sensitivity is 189 times as big as that of the commercial standard ultrasonic sensor. It is very interesting, especially it can be manufactured by a commercial 3D printer. Minor revision is suggested to improve the quality of this paper. 

1. The fabrication of the sensor with 3D printer is suggested to give more details.

2. The package structure for improving the thermal stability of the sensor is suggested to give more details.

Author Response

Dear Reviewer:

According to your comments, we have revised the manuscript. Please see the attachment.

Best regards,

Zongyu

Reviewer 2 Report

1. Lines 79-80: The XY resolution of the 3D printer (Formlabs, Form3) is 25 µm. Since in fig. 1a does not indicate the overall dimensions of the parts manufactured with its help, it is not clear whether this resolution of the 3D printer is sufficient.

2. Lines 87-88: The authors write that "Figure 2 shows the typical reflection spectrum of the sensor in air or underwater...". It is necessary to indicate whether fresh water or salty (sea) water was used in the experiment.

3. Lines 108-117: for the convenience of the reader, it may be worthwhile to provide a schematic representation of the sensor with all the geometric values ​​entered.

4. Lines 120-125: what grid was used in the simulation? How was the input action set?

5. Is it possible to make Figure 3 larger or enlarge one fragment, because arrow directions not clear?

6. Perhaps it would be better to provide a drawing showing the amount of membrane deformation?

7. The authors present results for different frequencies: line 151 "At 28.5 KHz and 50 KHz..." and line 163 "...18.5KHz and 200KHz". If this is not important, then for the convenience of the reader it is better to give the results for the same frequency values. Or you need to explain the selected frequency values.

8. Line 166: The authors report results for a temperature range of 30℃ - 60℃. It is required to explain why such a range was chosen (perhaps this is due to the potential scope of the sensor). It is also worth indicating at what temperature the research results given in the ms above were obtained (for example, in figures 2, 5, 6).

Author Response

Dear Reviewer:

We have revised the manuscript. Please see the attachment.

Best regards,

Zongyu
